# Host Immunosuppression Induced by *Steinernema feltiae*, an Entomopathogenic Nematode, through Inhibition of Eicosanoid Biosynthesis

**DOI:** 10.3390/insects11010033

**Published:** 2019-12-31

**Authors:** Miltan Chandra Roy, Dongwoon Lee, Yonggyun Kim

**Affiliations:** 1Department of Plant Medicals, Andong National University, Andong 36729, Korea; miltan.roy@yahoo.com; 2School of Environmental Ecology and Tourism, Kyungpook National University, Sangju 37224, Korea; whitegrub@knu.ac.kr

**Keywords:** *Steinernema feltiae*, eicosanoids, PLA_2_, *Plutella xylostella*, immunosuppression

## Abstract

*Steinernema feltiae* K1 (Filipjev) (Nematode: Steinernematidae), an entomopathogenic nematode, was isolated and identified based on its morphological and molecular diagnostic characteristics. Its infective juveniles (IJs) were highly pathogenic to three lepidopteran (LC_50_ = 23.7–25.0 IJs/larva) and one coleopteran (LC_50_ = 39.3 IJs/larva) insect species. Infected larvae of the diamondback moth, *Plutella xylostella* (L.) (Insecta: Lepidoptera), exhibited significant reduction in phospholipase A_2_ (PLA_2_) activity in their plasma. The decrease of PLA_2_ activity was followed by significant septicemia of the larvae infected with *S. feltiae*. Insecticidal activity induced by *S. feltiae* was explained by significant immunosuppression in cellular immune responses measured by hemocyte nodule formation and total hemocyte count (THC). Although *S. feltiae* infection suppressed nodule formation and THC in the larvae, an addition of arachidonic acid (AA, a catalytic product of PLA_2_) rescued these larvae from fatal immunosuppression. In contrast, an addition of dexamethasone (a specific PLA_2_ inhibitor) enhanced the nematode’s pathogenicity in a dose-dependent manner. To discriminate the immunosuppressive activity of a symbiotic bacterium (*Xenorhabdus bovienii* (Proteobacteria: Enterobacterales)) from the nematode, kanamycin was applied to after nematode infection. It significantly inhibited the bacterial growth in the hemolymph. Compared to nematode treatment alone, the addition of antibiotics to nematode infection partially rescued the immunosuppression measured by phenol oxidase activity. Consequently, treatment with antibiotics significantly rescued the larvae from the insecticidal activity of *S. feltiae*. These results suggest that immunosuppression induced by infection of *S. feltiae* depends on its symbiotic bacteria by inhibiting eicosanoid biosynthesis, resulting in significant insect mortality. However, the addition of antibiotics or AA could not completely rescue the virulence of the nematode, suggesting that the nematode itself also plays a role in its insecticidal activity.

## 1. Introduction

Insects exhibit cellular and humoral immune responses against various pathogens including microorganisms and multicellular parasites through their robust innate immune system [1]. Cellular immune responses use a hemocyte-spreading behavior to exhibit phagocytosis, nodule formation, and encapsulation [2]. In contrast, humoral immune responses represent chemical defenses using antimicrobial peptides (AMPs) and phenol oxidase (PO) [3,4]. PO catalyzes the melanin coat around encapsulated pathogens and produces chemically reactive quinones that are toxic to microbial pathogens [5]. PO activity is required for both cellular and humoral immune responses. Insects exert these immune functions soon after recognizing pathogen-associated molecular patterns by their pattern recognition receptors [6]. 

Entomopathogenic nematodes (EPNs) are especially promising because they possess various attributes that can allow them to be used as biological control agents for insect pests [7,8]. EPNs are classified into two genera: *Steinernema* and *Heterorhabditis*. *Steinernema* consists of at least 90 species including *S. affine*, *S. beddingi*, *S. carpocapsae*, *S. feltiae*, *S. glaseri*, *S. longicaudum*, and *S. monticolum. Heterorhabditis* consists of at least 20 species including *H. bacteriophora*, *H. indica*, and *H. megidis* [9]. *Steinernema* and *Heterorhabditis* EPNs possess symbiotic bacteria *Xenorhabdus* and *Photorhabdus*, respectively [10,11]. Generally, single nematode species can host one symbiotic bacterial species whereas each bacterial species can be hosted by more than one symbiotic nematode species [12]. Both EPN genera exhibit a converged evolutionary trait in their life cycle [13]. Infective juveniles (IJs) of EPNs live in the soil of diverse ecological systems and look for potential target insect hosts [14]. Once IJs infect a host through natural openings such as the mouth, anus, and spiracles, they can release symbiotic bacteria from nematode’s gut to insect hemolymph [15]. Bacteria can then suppress immune attacks of insect hosts to protect themselves and their symbiotic nematodes [16]. Under immunosuppressive conditions, these bacteria can multiply in the hemocoel and kill insects by septicemia or toxemia. Subsequently, nematodes can develop and reproduce in the insect cadaver [17]. 

To induce immunosuppression, symbiotic bacteria of EPNs can inhibit phospholipase A_2_ (PLA_2_) to shutdown eicosanoid biosynthesis of target insects [18]. Eicosanoids are oxygenated C20 polyunsaturated fatty acid (PUFA). They can be classified into three major groups: prostaglandins (PGs), leukotrienes (LTs), and epoxyeicosatrienoic acids (EETs) [19]. In vertebrates, arachidonic acid (AA) released from phospholipids (PLs) by PLA_2_ catalytic activity is usually a precursor for eicosanoid biosynthesis [20]. AA is then oxygenated into PGs by cyclooxygenase, LTs by lipoxygenase, and EETs by epoxidase activities of monooxygenases [21]. However, most terrestrial insects lack C20 PUFA presumably due to oxidative stress [22]. Instead, linoleic acid, which is rich in insect PLs, is released by PLA_2_ and changed into AA by desaturases and long chin fatty acid elongase [23]. Resulting PGs and LTs can mediate cellular immune responses by activating a hemocyte-spreading behavior [24] and humoral immune responses through activation of PO [25,26] and induction of AMP gene expression [27]. In mosquitoes, EETs are likely to mediate the humoral immunity in the midgut by elevating expression levels of some AMP genes [28]. The shutdown of eicosanoid biosynthesis by inhibiting PLA_2_ would prevent the synthesis of three different eicosanoids and lead to significant immunosuppression. This is one of pathogenicity strategies used by *Xenorhabdus* and *Photorhabdus* [16]. Thus, an addition of PLA_2_-catalytic product such as AA can significantly rescue bacterial pathogenicity [23,29]. However, such a rescue effect was little understood in insects infected with EPNs. If nematode contributes to the suppression of insect immune responses by targeting non-PLA_2_, AA addition would then incompletely rescue the immunosuppression induced by EPNs.

This study identified a new strain of *S. feltiae* and analyzed its pathogenicity by assessing insecticidal activity and immunosuppression via measuring PLA_2_ activity, counting total hemocyte count (THC) and phenoloxidase (PO) activity. It also tested a hypothesis that AA addition could rescue the immunosuppression induced by *S. feltiae* infection. Finally, it assessed the role of the nematode in suppressing immunosuppression by attenuating symbiotic bacteria by antibiotics.

## 2. Materials and Methods

### 2.1. Insect Rearing

Three different lepidopteran (*Spodoptera exigua* (Hűbner), *Maruca vitrata* (Fabricius), and *Plutella xylostella* (L.)) and one coleopteran (*Tenebrio molitor* L.) insects were used in this study. Larvae of *S. exigua* were collected from Welsh onion (*Allium fistulsum* L.) and reared on an artificial diet [30]. Larvae of *P. xylostella* were collected from cabbage (*Brassica rapa* L.) and reared on cabbage. Both *S. exigua* and *P. xylostella* were collected from Andong, Korea. Larvae of *M. vitrata* were collected from adzuki bean (*Vigna angularis* (Willd.) (Plantae: Fabales)) in Suwon, Korea and reared on an artificial diet [31]. Larvae of *T. molitor* were purchased from Bio Utility Institute, Inc. (Andong, Korea). Rearing conditions were: temperature of 25 ± 2 °C, photoperiod of 16:8 h (L:D), and relative humidity of 60 ± 5%. 

### 2.2. Chemicals

Arachidonic acid (AA: 5,8,11,14-eicosatetraenoic acid) and dexamethasone (DEX: (11β, 16α)-9-fluoro-11,17,21-trihydroxy-16-methylpregna-1,4-diene-3) were purchased from Sigma-Aldrich Korea (Seoul, Korea) and dissolved in dimethyl sulfoxide (DMSO). Anticoagulant buffer (ACB, pH 4.5) consisting of 186 mM NaCl, 17 mM Na_2_EDTA, and 41 mM citric acid was prepared. 

### 2.3. EPN Source and Culturing

An EPN isolate was obtained from a laboratory strain maintained in Nambo, Inc. (Jinju, Korea). The IJs were multiplied using fifth instar (L5) larvae of *S. exigua*. Briefly, approximately 200 IJs in 500 µL distilled water were applied to Petri dish (9 cm diameter, 3 cm height) containing an individual of L5 larva. Treated larvae were reared with an artificial diet for 3–5 days at rearing conditions. Brown-colored dead larvae were used to collect IJs using a White trap [32]. These collected IJs were then kept at 10 °C.

### 2.4. Molecular Identification of EPN Isolate

IJs (approximately 0.5 g) were used to extract genomic DNA (gDNA) using a method described by Kang et al. [33]. The internal transcribed spacer (ITS) region was amplified using primers GTTTTCCCAGTCACGACTTGATTACGTCCCTGCCCTTT and CAGGAAACAGCTATGACTTTCACTCGCCGTTACTAAGG reported by Vrain et al. [34]. Underlined sequences represent M13 forward and reverse sequences, respectively. For PCR amplification of the ITS region, the extracted gDNA (100 ng per 25 μL reaction volume) was used as a template. PCR consisted of 35 cycles of 1 min at 94 °C for denaturation, 1 min at 46 °C for annealing, and 1 min at 72 °C for extension. The PCR product was used for direct sequencing with M13 universal primers by Macrogen sequencing company (Seoul, Korea). Obtained nucleotide sequences were analyzed using BlastN program of the National Center for Biotechnology Information (www.ncbi.nlm.nih.gov). Phylogenetic tree analysis was performed using a Neighbor-Joining method with MEGA6 [35]. Bootstrap values on branches were estimated with 1000 repetitions.

### 2.5. Virulence Assay of EPN

Insecticidal activity of the EPN isolate was determined by applying different concentrations (0–320 IJs/larva) to filter paper in Petri dish containing 10 larvae. The experimental unit (=Petri dish) was replicated three times per nematode concentration. Third instar larvae of three lepidopteran species (*S. exigua*, *M. vitrata*, and *P. xylostella*) were used. Larvae of a coleopteran insect (*T. molitor*) with body length of approximately 2 cm were also used in the assay. Treated larvae were provided with diets and kept in rearing conditions. Median lethal concentration (LC_50_) was estimated from the mortality data at 72 h after treatment. Median lethal time (LT_50_) was estimated after treatment with an IJ concentration (80 IJs/larva). To estimate LT_50_, mortality was assessed every 8 h for 3 days after nematode infection. LC_50_ and LT_50_ values were estimated using EPA Probit program, version 1.5 (U.S. Environmental Protection Agency, Cincinnati, OH, USA).

### 2.6. Assessment of Nodule Formation after Nematode Infection

For nodulation assay, *Escherichia coli* Top10 cells were used. These bacteria were heat-killed at 95 °C for 10 min. Fourth instar larvae of *P. xylostella* were infected with 80 IJs/larva. After nematode infection, larvae were randomly selected every 4 h and injected with 1.95 × 10^4^ cells/larva using a Sutter CO_2_ picopump injector (PV830, World Precision Instrument, Sarasota, FL, USA). Control larvae without nematode infection were also injected with bacteria at the same time periods for 24 h. At 8 h after bacterial injection, nodules were counted by dissecting larvae under a stereomicroscope (Stemi SV 11, Zeiss, Jena, Germany) at 50× magnification. For each time point, 10 larvae were used in each of control and nematode treatments.

To assess effects of AA or DEX on nodule formation in larvae infected with *S. feltiae*, at 24 h after treatment with nematode (80 IJs/larva), larvae were injected with 10 µg of AA or DEX along with bacterial injection. At 8 h after bacterial injection, the number of nodules formed was counted as described above. 

### 2.7. Bioassay of *P. xylostella* Larvae Infected with S. feltiae with Addition of DEX or AA

For nematode treatment, 80 IJs/larva were used in the Petri dish assay (10 larvae of *P. xylostella* per dish). Just before nematode treatment, test larvae were injected with DEX or AA at different doses [0 (DMSO only), 0.1, 1, 10, and 100 µg]. Mortality was assessed at 72 h after nematode infection. Each treatment consisted of three replicates. 

### 2.8. Total Hemocyte Count (THC) Analysis after Nematode Treatment

Fourth instar larvae of *P. xylostella* were treated with the nematode isolate (80 IJs/larva). Hemolymph was then collected into ACB by cutting a proleg on the abdomen and aspirating the exuded hemolymph with glass capillaries (TW100-4, World Precision Instrument, Sarasota, FL, USA) at 4 h intervals for 24 h. Hemocytes were counted with a hemocytometer (Neubauer improved bright line, Superior Marienfield, Lauda-Königshofen, Germany) under a phase contrast microscope (BX41, Olympus, Tokyo, Japan). Each treatment was independently replicated three times.

To evaluate effect of AA on THC, test larvae were injected with AA or DEX at a dose of 10 µg of just before nematode treatment (80 IJs/larva). Hemolymph was collected at 16 h after nematode treatment and assessed for THC as described above. 

### 2.9. Antibiotic Screening against *X. bovienii*

Three different antibiotics (ampicillin, tetracycline, and kanamycin from Sigma-Aldrich Korea) were prepared at 0, 0.01, 0.1, 10, and 100 μg/µL. A symbiotic bacterium, *X. bovienii* SS 2004 (Proteobacteria: Enterobacterales), was kindly provided by Dr. Sophie Gaudriault (INRA, Monfelliae, France). The bacterium was cultured in Luria-Bertani (LB) medium at 28 °C. Thirty µL of an overnight grown bacterial suspension was plated onto LB agar medium and left for 5 min under a clean bench. Filter paper discs (10 mm diameter) impregnated with different antibiotics were placed on the agar surface. After 16 h of culture at 28 °C, the clear zone around the disc was measured in the distance from the edge of the disc. Each dose treatment was replicated three times.

### 2.10. Effect of Kanamycin in Suppressing Bacterial Growth in *P. xylostella* Larvae Infected with *S. feltiae*


To determine the optimal concentration of kanamycin for injection, different kanamycin concentrations were injected to fourth instar larvae of *P. xylostella*. Mortality was measured at 72 h after treatment. Each treatment used 10 larvae and replicated three times. 

Larvae were then injected with 0.1 µg kanamycin and treated with nematode *S. feltiae* at 80 IJs/larvae. At 8 h after the nematode infection, hemolymph was collected and plated onto LB plate to count bacterial colonies after culturing at 28 °C for 16 h. Each treatment was replicated three times. For each replication, 10 larvae were used. 

### 2.11. Phenol Oxidase (PO) Enzyme Activity 

PO activity was determined using 3,4-dihydroxy-L-phenylalanine (DOPA, Sigma-Aldrich Korea) as a substrate. DOPA was dissolved in 100 mM phosphate-buffered saline (PBS, pH 7.4). Fourth instar larvae of *P. xylostella* larvae were infected with nematodes at 80 IJs/larva. PO activity was measured at 8 h after the nematode infection. Hemolymph samples were collected from ~30 treated larvae as described above and centrifuged at 4 °C for 5 min at 800× *g* to obtain supernatant plasma samples. Each reaction volume (200 µL) for PO activity measurement consisted of 180 µL of 10 mM DOPA and 20 µL of plasma sample. Absorbance was read at 495 nm using a VICTOR multi label Plate reader (PerkinElmer, Waltham, MA, USA). PO activity was expressed as ΔABS/min/µL plasma. Each treatment consisted of three biologically independent replicates. 

To determine the effect of kanamycin on PO activity of nematode-infected larvae, test larvae were injected with 0.1 µg of kanamycin before nematode infection. Untreated naive larvae were also analyzed. PO activity was analyzed at the same time interval as described above.

### 2.12. PLA_2_ Activity Measurement in Plasma after *S. feltiae* Infection

Fourth instar larvae of *P. xylostella* were used in PLA_2_ enzyme assay. For nematode treatment 80 IJs per larva were used by spraying on filter paper in a Petri dish. Control used water spray. At different time points, the hemolymph was collected as described above. These collected hemolymph samples were then centrifuged at 800× *g* for 3 min. The supernatant was used as a plasma sample. Each plasma sample was obtained from 10 larvae. Each treatment was replicated three times. To measure PLA_2_ activity of the collected plasma sample, a commercial assay kit (sPLA_2_ Assay Kit, Cayman Chemical, Ann Arbor, MI, USA) was used with diheptanoyl thiophosphatidylcholine as a substrate for the enzyme. Protein concentration was determined by Bradford [36] assay using bovine serum albumin as standard. Catalyzed product was then allowed to react with Ellman’s reagent (5,5′-dithio-bis-(2-nitrobenzoic acid), DTNB) to create 5-thio-2-nitrobenzoic acid, a colored product. DTNB was prepared in 10 mM in 0.4 M Tris buffer (pH 8.0). Assay buffer was 25 mM Tris (pH 7.5) containing 10 mM CaCl_2_, 100 mM KCl, and 0.3 mM Triton X-100. A reaction volume of 175 μL contained 10 μL plasma sample, 10 μL DTNB, 5 μL assay buffer, and 150 μL of 1.66 mM substrate. As a negative control, the same volume of reaction mixture consisted of 10 μL DTNB, 15 μL assay buffer, and 150 μL substrate. Absorbance changes at 405 nm were measured and plotted to obtain the slope of a linear portion of the curve. Absorbance for non-enzymatic blank control was calculated and subtracted from sample wells. The actual extinction coefficient for DTNB was 10.66 mM^−1^cm^−1^.

### 2.13. Statistical Analysis

Mortality data were transformed by arsine for analysis of variance (ANOVA). Means and variances of treatments were analyzed by one-way ANOVA using PROC GLM of SAS program [37]. All means were compared by least squared difference (LSD) test at Type I error = 0.05.

## 3. Results

### 3.1. Identification of a Nematode Isolate

IJs of the nematode isolates were infective to larvae of *P. xylostella*. Larvae were killed by the nematode, becoming brown-colored cadavers (Figure 1A). The nematode infection reproduced several hundreds of subsequent IJs (Figure 1B). In cadavers, different developmental stages of nematodes were observed (Figure 1C–G).

To identify the nematode isolate, ITS regions were amplified and sequenced (Figure 2A). A total of 950 bp was sequenced for the isolate and compared with a known sequence of *Caenorhabditis elegans* (Nematoda: Rhabditida) [38] to confirm that these sequenced regions included ITS. The determined sequence contained 5.8S rRNA, ITS-1, and ITS-2 regions as well as partial sequences of 18S and 28S rRNAs. Its sequence was deposited at GenBank with an accession number of MN093395.1. The sequence showed high similarities (>99%) with known ITS sequences of several *S. feltiae* strains (Table 1). To evaluate the relationship between the isolate and other known *Steinernema* spp., a phylogenetic tree was constructed using ITS sequences (Figure 2B). The isolate was significantly (bootstrap value = 100) clustered with a local isolate of *S. feltiae* from Czech Republic [39] among 25 different species of *Steinernema*.

Molecular identification was further supported by morphometric characteristics of the isolate. All morphological characteristics were obtained from IJ stage, including total body length and width (Figure 1C), esophagus length (Figure 1D), and tail length (Figure 1E). Sizes or ratios of these morphological characteristics were within variations of those in known *S. feltiae* species (Table 2). The isolate was identified as *S. feltiae* K1.

### 3.2. Insecticidal Activity of *S. feltiae* K1

*S. feltiae* K1 was virulent against larvae of three lepidopteran (*P. xylostella, M. vitrata*, and *S. exigua*) species and one coleopteran (*T. molitor*) species (Figure 3). Different concentrations of IJ treatment resulted in significant differences in mortality of these four species of insects (*F* = 246.74; df = 6, 56; *p* < 0.0001). Susceptibilities of these insect species to the nematode were slightly different, with *T. molitor* being the least susceptible to infection (*F* = 3.49; df = 3, 56; *p* = 0.0215). Also, estimates of LC_50_ indicated that *S. exigua*, *P. xylostella*, and *M. vitrata* larvae were much more susceptible to the nematode than *T. molitor* larvae (Table 3). However, there was little significant difference in the speed-to-kill of *S. feltiae* K1 against these four insect species based on similar LT_50_s.

### 3.3. Insecticidal Virulence of *S. feltiae* K1 by Down-Regulating PLA_2_ Activity

After *P. xylostella* larvae were infected with *S. feltiae* K1, PLA_2_ activity changes in the plasma of infected larvae were monitored for 24 h (Figure 4A). The nematode infection significantly (*p* < 0.05) inhibited plasma PLA_2_ activity. The inhibition began at 8 h after treatment with the nematode. At 24 h after the treatment, some infected larvae began to die. In contrast, untreated (naive) larvae did not show any significant change in PLA_2_ activity during the test period.

To see the effect of PLA_2_ inhibition on insecticidal activity of the nematode, dexamethasone (‘DEX’, a PLA_2_ inhibitor) was used for treatment. It further significantly (*p* < 0.05) increased the virulence of the nematode against *P. xylostella* larvae (Figure 4B). In contrast, addition of arachidonic acid (‘AA’, a catalytic product of PLA_2_) to larvae infected with *S. feltiae* K1 at dose of 100 μg/larva significantly (*p* < 0.05) rescued these larvae from the lethality.

### 3.4. Infection of *S. feltiae* K1 Resulted in Immunosuppression and Cytotoxicity

The inhibition of PLA_2_ by the infection of *S. feltiae* K1 suggested that there might be an immunosuppression due to the lack of eicosanoid immune mediators [16]. To test this hypothesis, a cellular immune response measured by nodule formation was monitored for 24 h after infection with the nematode (Figure 5A). The infection with the nematode significantly (*p* < 0.05) suppressed nodule formation. The suppression began at 4 h after infection with the nematode. At 24 h post-infection, larvae infected with *S. feltiae* K1 almost lost their nodule formation ability. To test a hypothesis that the immunosuppression might be induced by downregulated PLA_2_ activity inhibited by the nematode, DEX or AA was added to nematode-infected larvae (Figure 5B). DEX treatment further significantly (*p* < 0.05) suppressed nodule formation while AA addition significantly (*p* < 0.05) rescued such immunosuppression.

To explain the insecticidal activity of *S. feltiae* K1, cytotoxicity of the nematode infection was assessed in hemocytes (Figure 6A). *P. xylostella* larvae had over 10^6^ hemocytes per mL of hemolymph. With increase of infection period, *S. feltiae* K1 significantly (*p* < 0.05) induced cytotoxicity against hemocytes of *P. xylostella*. Significant cytotoxicity began at 4 h post-infection. At 24 h, infected larvae lost more than 70% hemocytes and had only 3 × 10^5^ hemocytes per mL of hemolymph. To explain the cytotoxicity caused by the inhibitory activity of *S. feltiae* K1 against PLA_2_, DEX, or AA was added to nematode-infected larvae (Figure 6B). DEX treatment significantly (*p* < 0.05) potentiated the cytotoxicity of nematode while AA addition significantly (*p* < 0.05) rescued hemocytes.

### 3.5. Role of Symbiotic Bacteria in Insecticidal Activity of *S. feltiae* K1

*S. feltiae* possesses species-specific symbiotic bacterium *X. bovienii* in the intestine [41]. This suggests that the insecticidal activity of *S. feltiae* K1 might be induced by both symbiotic bacteria and their host, the nematode. To discriminate bacterial contribution from the nematode, treatment with antibiotics was performed after nematode infection. First, we selected an effective antibiotic against *X. bovienii* based on an inhibition zone assay (Figure 7A). Among three antibiotics used, kanamycin was the most effective one in suppressing the bacterial growth at all test concentrations. We then treated *P. xylostella* larvae with different concentrations of kanamycin (Appendix A). At a high concentration (100 μg/larva), kanamycin was harmful for the survival of *P. xylostella* larvae. When a relatively low concentration (0.1 μg/larva) of kanamycin was injected to larvae already treated with *S. feltiae* K1, bacterial growth was suppressed by more than 75% (Figure 7B). Under these antibacterial treatment conditions, contributions of bacteria and nematode to insecticidal activity against *P. xylostella* were separately assessed (Figure 7C). Kanamycin treatment along with nematode infection significantly (*p* < 0.05) decreased the insecticidal activity induced by infection of *S. feltiae* K1 alone. However, it did not completely rescue nematode-infected larvae.

PO activity of *P. xylostella*, which would be required for both cellular and humoral immune responses, was significantly (*p* < 0.05) suppressed by the injection with live *X. bovienii* bacteria while such activity was significantly enhanced by the injection with heat-killed *X. bovienii* bacteria (Figure 7D). The nematode infection also significantly suppressed PO activity. However, an addition of kanamycin partially rescued the PO activity suppressed by the nematode infection.

## 4. Discussion

EPN infection can lead to septicemia of target insects. Such septicemia is induced by immunosuppression [29]. Eicosanoids play a crucial role in mediating cellular and humoral immune responses in insects [16]. Both symbiotic bacteria (*Xenorhabdus* and *Photorhabdus*) of EPNs can inhibit PLA_2_ to shut down eicosanoid biosynthesis of target insects [18]. Indeed, Hasan et al. [23] have demonstrated a positive correlation between the degree of PLA_2_ inhibition by EPNs and their insecticidal activities. However, the role of EPN without symbiotic bacteria in killing target insects by immunosuppression was unclear. This study analyzed independent roles of bacteria and nematodes in inducing immunosuppression and insecticidal activity ultimately.

A new isolate of *S. feltiae* K1 was identified based on molecular and morphological characteristics. ITS sequence and IJ morphological characteristics indicated that the nematode isolate was a strain of *S. feltiae*. ITS sequences are powerful for identifying EPNs without morphological characteristics. For example, in an ecological analysis of EPNs and their symbiotic bacteria in Thailand, ITS sequences alone have been used to identify EPNs collected from soil samples [42]. This current study supports the usefulness of ITS sequence for EPN identification because morphological characteristics clearly support the molecular identification.

Infection by *S. feltiae* K1 had significant lethal effects on four different insect species (three lepidopteran species and one coleopteran species). There was a variation in insecticidal activities among insect species. Especially, the coleopteran species, *T. molitor*, was less susceptible to the nematode infection than the three lepidopteran species. Compared to other similar nematode species [23], *S. feltiae* K1 was more potent (more than two times based on LC_50_ values) than *S. carpocapsae* against the same host under the same assay conditions. However, LT_50_s (= speed-to-killing insects) of *S. feltiae* showed some variations against different insect species. This indicates that the insecticidal activity of *S. feltiae* K1 is not determined by IJ density only if we knew the number of the infected IJs within the target insects. Symbiotic bacteria might also play a role in the insecticidal activity of the nematode. Jung and Kim [43] have measured the speed-to-kill of *X. nematophila* (Proteobacteria: Enterobacterales) is 16 h against *P. xylostella* larvae. Our current LT_50_s ranged from 46 to 50 h post-infection. Although bacterial species are different, at least 30 h might be necessary to reach the lethal level of bacterial population required to cause lethal septicemia. Thus, the initial 30 h post-infection would be a struggling period between host (insect) defense and virulence of the nematode/bacterial complex. The pathogen–insect interaction involved in the immunosuppression induced by the nematode/bacterial complex was further analyzed in the following section.

Cellular immune response of *P. xylostella* was severely impaired by the infection of *S. feltiae* K1. More than 20 nodules were formed in each larva of *P. xylostella* in response to nonpathogenic bacterial infection. However, the nematode infection progressively suppressed the cellular immune response and led to one or two nodules per larva in response to the same number of bacteria used for challenge at 24 h post-infection. With temporal decrease of cellular immune response, there was a similar decrease of PLA_2_ activity in the plasma of *P. xylostella* after the nematode infection. PLA_2_ activity in the plasma is due to a secretory type of PLA_2_ (sPLA_2_) in *Spodoptera exigua* [44]. Our current PLA_2_ activity was measured using sPLA_2_ substrate. Thus, the decrease of PLA_2_ activity in the plasma of *P. xylostella* can be explained by the inhibition of sPLA_2_ of *S. feltiae* K1. The sPLA_2_ of *S. exigua* plays a crucial role in eicosanoid biosynthesis. It also exhibits a direct antibacterial activity like other antimicrobial peptides [44]. Thus, inhibition of plasma PLA_2_ activity should be required for the pathogenicity of *S. feltiae* K1 to protect the symbiotic bacteria. In addition, eicosanoid biosynthesis can be inhibited by suppressing PLA_2_ activity. Eicosanoids are known to mediate nodule formation of hemocytes by activating the hemocyte-spreading behavior via stimulation of actin polymerization factors [2]. After infection with *S. feltiae* K1, PLA_2_ activity and nodulation were also downregulated. Hemolysis accompanying decrease of THC was induced by the infection of *S. feltiae* K1. Interestingly, the cytotoxicity of *S. feltiae* K1 could be manipulated by the addition of DEX or AA. This suggests that the cytotoxicity of *S. feltiae* K1 is influenced by eicosanoids. Cho and Kim [45] have reported that *Xenorhabdus* and *Photorhabdus* are cytotoxic to insect hemocytes via apoptosis. Hemocytes of the larvae infected with these bacteria exhibited membrane blebbing, nuclear chromatin condensation, and DNA fragmentation [45]. This suggests that *S. feltiae* K1 can release its symbiotic bacteria that can then synthesize and release apoptosis-inducing factor(s) to kill hemocytes. Indeed, *X. bovienii*, a symbiotic bacterium of *S. feltiae*, can synthesize cytotoxic xenocycloin compounds [46]. Thus, the regulation of cytotoxicity by DEX or AA can be explained by the control of immune responses of *P. xylostella* which manipulates selective pressure on bacterial survival and subsequent xenocycloin production level.

Effective treatment with antibiotics against *X. bovienii* rescued the larvae infected with *S. feltiae* K1. Kanamycin was the most effective one in preventing the growth of *X. bovienii*. Indeed, treatment with antibiotics suppressed the bacterial population in the hemolymph of *P. xylostella* larvae after infection with *S. feltiae* K1. Under this condition, *S. feltiae* K1 partially lost its insecticidal activity. This clearly exhibits the insecticidal role of symbiotic bacteria. There is a variation in insecticidal activity among *X. bovienii* strains, in which the composition of functional genome associated with virulence is different [47]. Interestingly, treatment with antibiotics to kill symbiotic bacteria did not completely rescue the larvae infected by *S. feltiae* K1. This suggests that the nematode itself possesses insecticidal activity without the presence of symbiotic bacteria. Furthermore, PO analysis indicated that treating larvae infected by the nematode with antibiotics could still significantly suppress the induction of the enzyme activity. These results suggest that the contribution of nematode to immunosuppression is distinct from that of the symbiotic bacteria. CS03 strain of *X. bovienii* is a non-virulent strain. However, it exhibits low virulence when it is associated with its *Steinernema* symbiont [48]. This supports the role of the nematode in insecticidal activity without the presence of symbiotic bacteria. Metabolite analysis of *S. feltiae* IJs using mass spectrometry identified 266 proteins, among which 52 proteins were conserved among nematode metabolites known to be associated with tissue-damaging and immune-modulating proteins [49]. In addition, the nematode cuticle plays a crucial role in avoiding insect immune surveillance by disguising nematode surface [50]. Thus, several chemical and physical factors derived from *S. feltiae* K1 might be responsible for its immunosuppression and insecticidal activity. However, we cannot avoid the pathogenic contribution of the gut microbiota in the host insects, which might infect the insect hemocoel during penetration of IJs from gut lumen. To clarify the role of nematode on the immunosuppression, use of axenic IJs might be optimal in a future study rather than use of antibiotics to kill the symbiotic bacteria.

## 5. Conclusions

This study identified a nematode isolate to be *S. feltiae* K1 based on its morphological and molecular characteristics. *S. feltiae* K1 was highly pathogenic to lepidopteran and coleopteran insect pests. Its pathogenicity was dependent on the inhibition of target insect PLA_2_ activity because addition of AA, an enzyme catalytic product, significantly rescued the virulence of *S. feltiae* K1. Treatment with an antibiotic to inhibit the growth of symbiotic bacteria significantly rescued nematode-infected insects. However, this study indicated a pathogenic role of the nematode in suppressing insect immune response. It might be independent of the contribution of its symbiotic bacteria.

## Figures and Tables

**Figure 1 insects-11-00033-f001:**
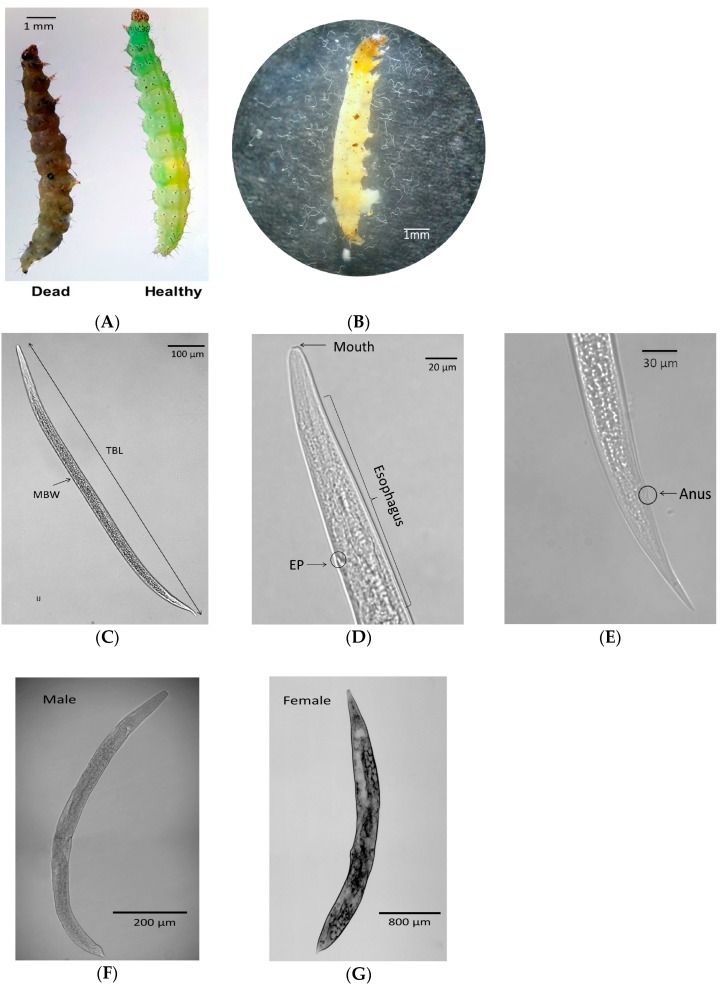
Morphological structures of *S. feltiae* isolate at different developmental stages. (**A**) Dead symptom of *P. xylostella* larva infected with the nematode isolate. (**B**) Multiplied nematodes released from the insect cadaver. (**C**) Whole body shape of the isolate infective juvenile (IJ). (**D**) IJ esophagus and excretory pore in circle. (**E**) IJ tail showing anus in circle. (**F**) Whole body of an adult male. (**G**) Whole body of an adult female.

**Figure 2 insects-11-00033-f002:**
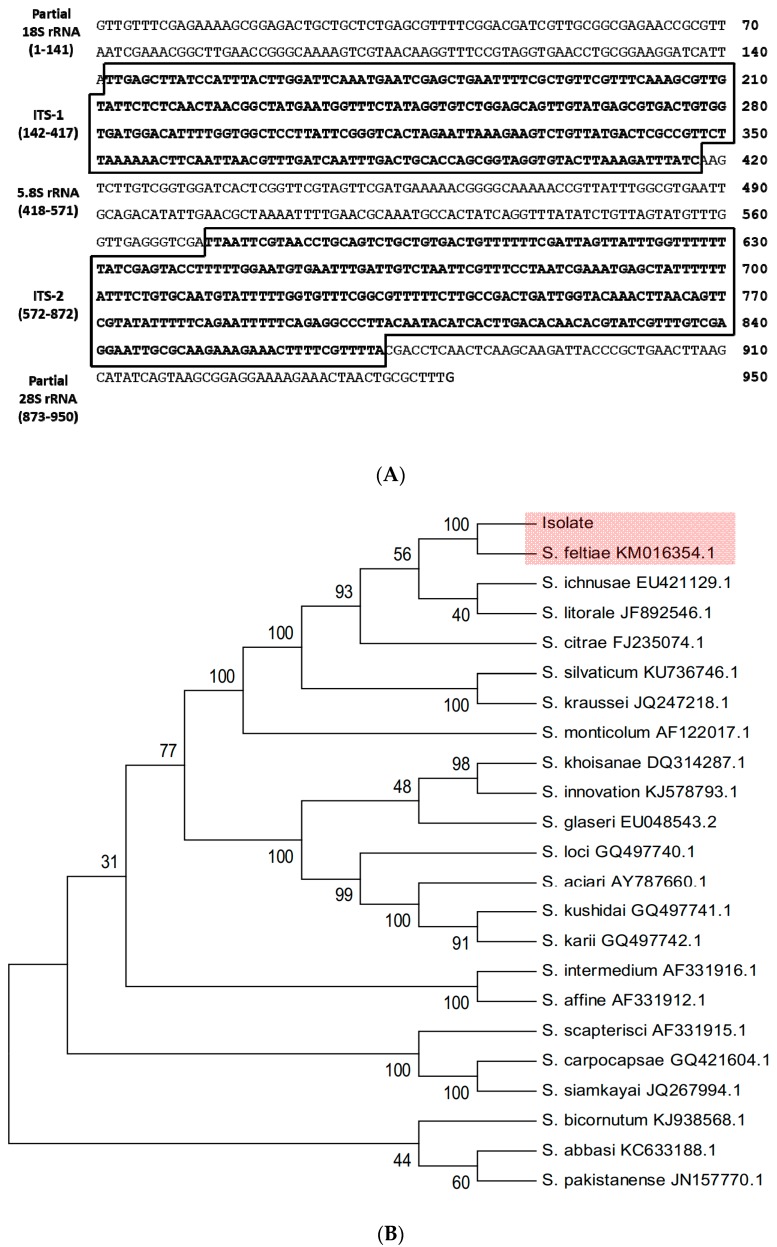
Molecular identification of *S. felt**iae* isolate. (**A**) Sequence analysis of ITS/rRNA region of the nematode isolate. Two ITS regions are marked with boxes. The DNA sequence was deposited to GenBank with an accession number of MN093395.1. (**B**) Phylogenetic relationship of the nematode isolate with other *Steinernema* species based on ITS and rRNA sequences. The tree was constructed with the Neighbor-joining method using MEGA6.0. Bootstrapping values on branches were obtained with 1000 repetitions. GenBank accession numbers of ITS/rRNA sequence are followed by species names.

**Figure 3 insects-11-00033-f003:**
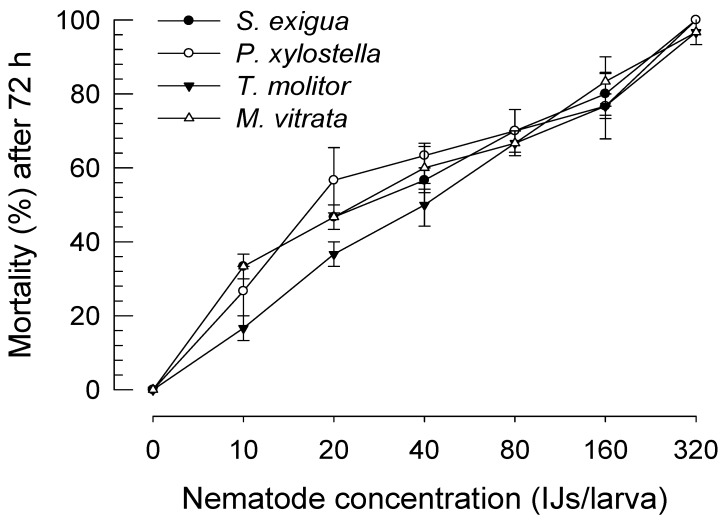
Virulence of *S. feltiae* K1 to larvae of four target insects: *S. exigua, T. molitor, Maruca vitrata*, and *P. xylostella*. Infective juveniles (IJs) at different concentrations were applied to a filter paper in a Petri dish (9 cm in diameter, 3 cm in height) containing 10 larvae of target insects. Each treatment was replicated three times. Nematode dose was calculated by dividing the applied IJ number by 10 larvae to get IJs per larva. Mortality was recorded at 72 h after IJ treatment. Dead insects were dissected to determine whether the lethality was due to the nematode infection.

**Figure 4 insects-11-00033-f004:**
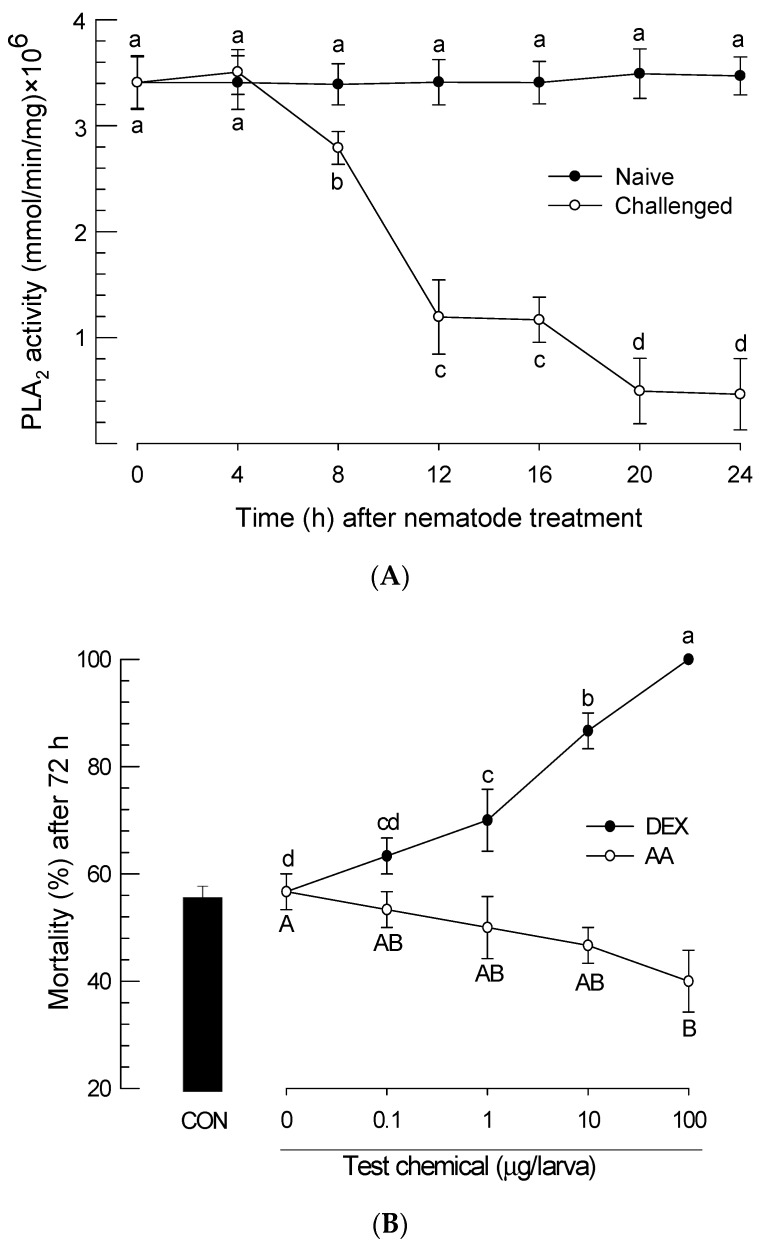
Effect of PLA_2_ inhibition induced by *S. feltiae* K1 infection on insecticidal activity against *P. xylostella* larvae. (**A**) Inhibition of PLA_2_ activity in the plasma of *P. xylostella* larvae by the nematode infection (‘Challenged’) of *S. feltiae* K1. Hemolymph was collected every 4 h after the nematode infection to prepare plasma samples. Each treatment was replicated three times. (**B**) Influence of arachidonic acid (‘AA’, 10 μg/larva) or dexamethasone (‘DEX’, 10 μg/larva) addition on mortality of *P. xylostella* larvae infected with *S. feltiae* K1. Indicated concentrations of test chemicals were injected into hemocoel of test insects followed by nematode treatment (80 IJs/larva). Subsequent mortality was measured at 72 h after nematode treatment. Each treatment was replicated three times. For each replication, 10 larvae were used. Different letters above standard error bars indicate significant difference among means at Type I error = 0.05 (LSD test).

**Figure 5 insects-11-00033-f005:**
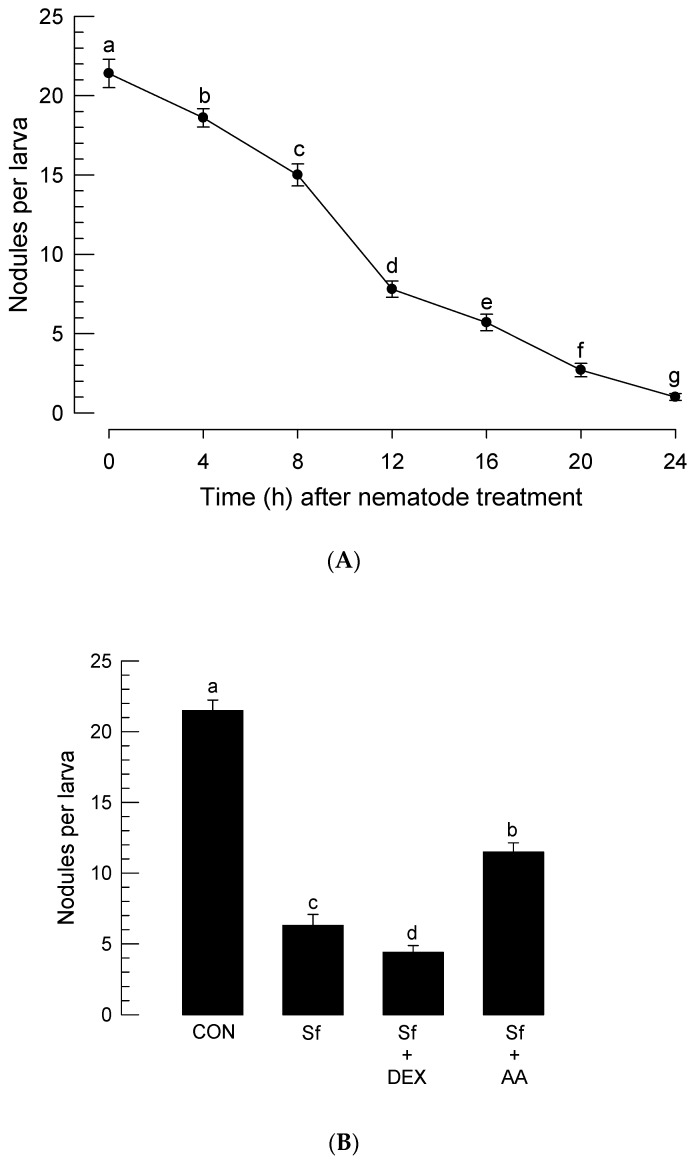
Suppression of hemocyte nodule formation in *P. xylostella* larvae infected with *S. feltiae* K1. (**A**) Temporal change in immune response after nematode infection. Fourth instar larvae of *P. xylostella* were treated with 80 IJs per larva. At different time points, these treated larvae were injected with *E. coli* (1.86 × 10^4^ cells/larva). Nodules were counted at 8 h after the bacterial treatment. (**B**) Effect of arachidonic acid (‘AA’, 10 μg/larva) or dexamethasone (‘DEX’, 10 μg/larva) addition on nodulation of *P. xylostella* larvae infected with *S. feltiae* K1 (‘Sf’). Each treatment was replicated three times and each replication used 10 individual larvae per treatment. Different letters above standard error bars indicate significant difference among means at Type I error = 0.05 (LSD test).

**Figure 6 insects-11-00033-f006:**
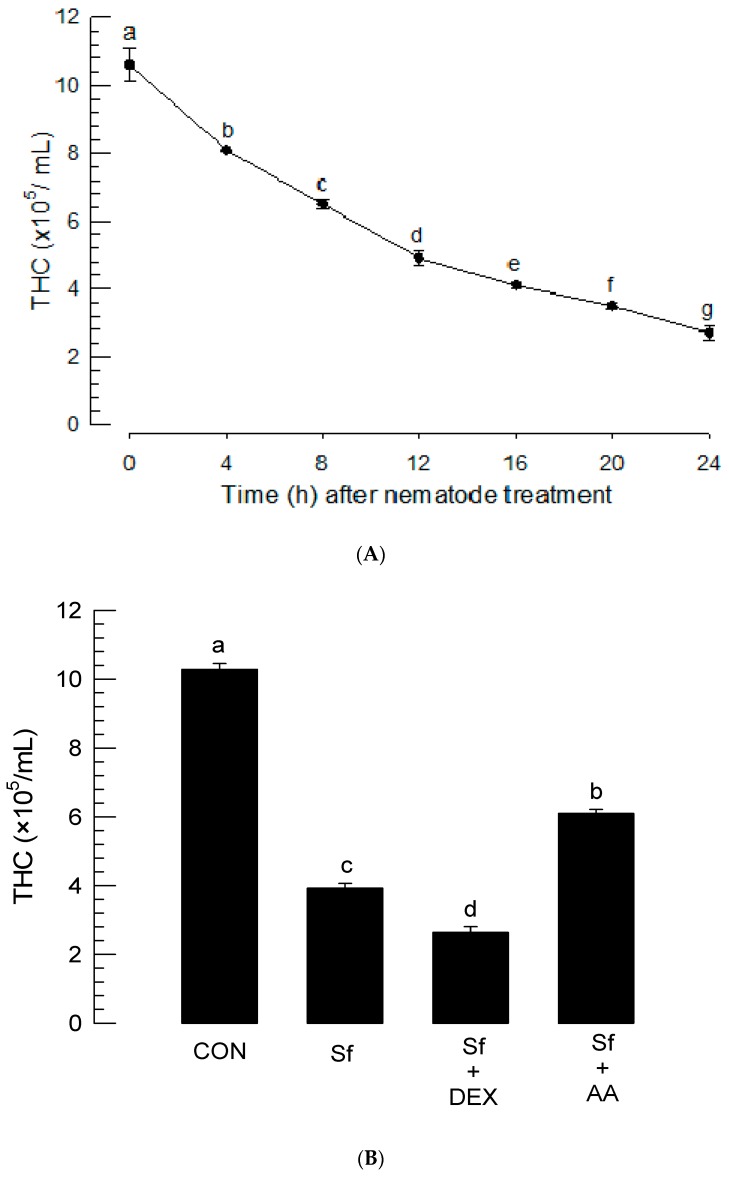
Suppression of total hemocyte counts (THC) of *P. xylostella* larvae infected with *S. feltiae* K1. (**A**) Temporal change of THC after the nematode infection. Fourth instar larvae of *P. xylostella* were treated with 80 IJs per larva. (**B**) Rescue effect of arachidonic acid (‘AA’, 10 μg/larva) or dexamethasone (‘DEX’, 10 μg/larva) addition on THC. Test chemicals were injected just before nematode (‘Sf’) treatment (80 IJs/larva). THC was assessed at 16 h after the nematode treatment. Each treatment was replicated three times. Different letters above standard error bars indicate significant difference among means at Type I error = 0.05 (LSD test).

**Figure 7 insects-11-00033-f007:**
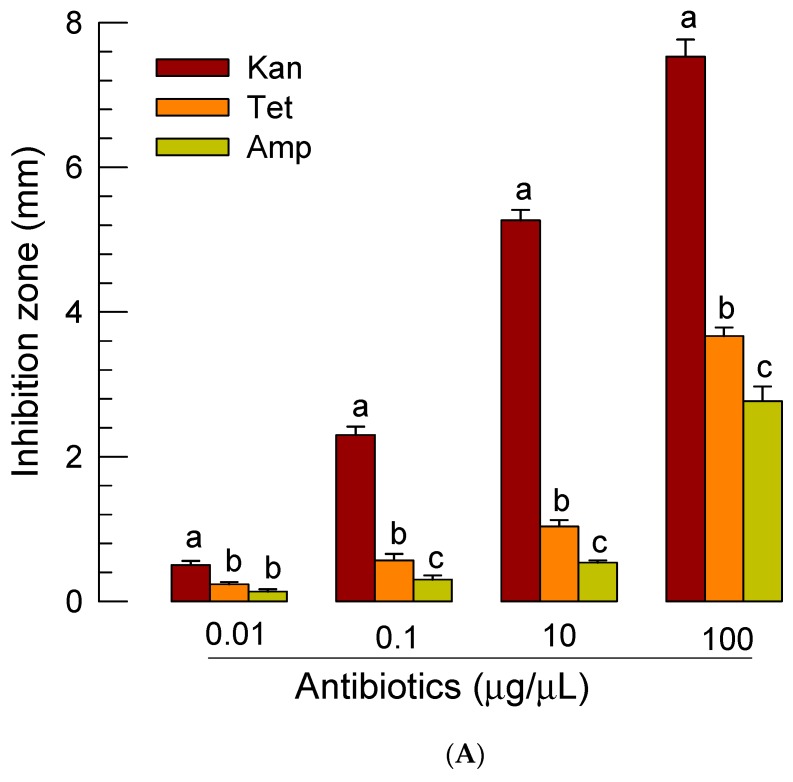
Independent roles of nematode and its symbiotic bacteria of *S. feltiae* K1 (‘Sf’) in their pathogenicity against *P. xylostella*. (**A**) Screening effective antibiotics against *X. bovienii* SS2004 growth using inhibition zone assay. Kanamycin (‘Kan’), tetracyclin (‘Tet’), and ampicillin (‘Amp’) were screened. (**B**) In vivo effect of Kan (0.1 μg/larva) on bacterial growth in *P. xylostella* larvae treated with 80 IJs of *S. feltiae* K1 per larva. Antibiotics treatment was performed just before the nematode treatment. Bacterial colonies were counted at 8 h after nematode treatment and expressed as colony forming units (cfu) per mL of hemolymph. Photos above the graph demonstrate bacterial colonies in each treatment. (**C**) Rescue effect of Kan (0.1 μg/larva) on larvae treated with 80 IJs of *S. feltiae* K1 per larva. Two different concentrations of *X. bovienii* (‘Xb’) were injected to hemocoel of *P. xylostella* larvae. Each treatment was replicated three times. Each replication used 10 larvae. (**D**) Rescue effect of Kan (0.1 μg/larva) on phenoloxidase (PO) activity of the larvae treated with 80 IJs of *S. feltiae* K1 per larva. Heat-killed *X. bovienii* (‘HK-Xb’) was injected to larvae at a concentration of 1000 cells per larva. PO activity was measured at 8 h post infection. Each treatment was replicated three times. Different letters above standard error bars indicate significant difference among means at Type I error = 0.05 (LSD test).

**Table 1 insects-11-00033-t001:** Identification of an EPN isolate using ITS sequence. The sequence region (950 nucleotide (nt)) contains partial 18S rRNA (141 nt), ITS1 (276 nt), 5.8S rRNA (154 nt), ITS2 (301 nt), and partial 28S rRNA (78 nt).

Species Blasted in NCBI-GenBank	GenBank Accession Number	Total Score	E Value	Identity (%)
*S. feltiae* strain 626	KM016348.1	1724	0.0	99.68
*S. feltiae* isolate WG-01	MK294325.1	1718	0.0	99.58
*S. feltiae* strain SN	AF121050.2	1718	0.0	99.58
*S. feltiae* strain T 92	AY230185.1	1714	0.0	99.47
*S. feltiae* strain 626	KM016351.1	1714	0.0	99.47

**Table 2 insects-11-00033-t002:** Identification of *S. feltiae* isolate based on morphological characteristics of infective juveniles.

Morphometric Characters	*S. feltae* Isolate Mean ± SE (n = 20)	*S. feltae* ^1^
TBL (total body length), µm	805 ± 10.6	849 (736–950)
MBW (maximum body width), µm	26.8 ± 0.6	26 (22–29)
EP (excretory pore), µm	62.0 ± 1.5	62 (53–67)
ES (esophagus length), µm	138 ± 2.4	136 (115–150)
TL (tail length), µm	83.9 ± 1.7	81 (70–92)
TBL/MBW	30.3 ± 0.8	31 (29–33)
TBL/ES	5.9 ± 0.1	6.0 (5.3–6.4)
TBL/TL	9.6 ± 0.2	10.4 (9.2–12.6)
EP/ES × 100	45.1 ± 1.3	45 (42–51)
EP/TL × 100	74.2 ± 2.2	78 (69–86)

^1^ Nguyen and Smart [40].

**Table 3 insects-11-00033-t003:** Insecticidal activities of *S. feltiae* K1 against four different insect species. Third instar larvae of three lepidopteran species were used in bioassays. Larvae of a coleopteran species, *T. molitor*, with body length of approximately 2 cm were used in bioassays. To estimate the median lethal concentrations (LC_50_s), test nematode concentrations were 0, 10, 20, 40, 80, 160, and 320 IJs per larva. An experimental unit (=each Petri dish) contained 10 larvae. It was replicated three times per nematode concentration. Median lethal times (LT_50_s) were estimated as nematode concentration of 80 IJs per larva.

Target Insects	N	LC_50_ (IJs/Larva)	Slope ± SE	LT_50_ (h)	Slope ± SE
*S. exigua*	30	24.85 (11.9 ± 51.6)	1.08 ± 0.16	46.36 (39.0 ± 55.1)	4.15 ± 0.04
*T. molitor*	30	39.33 (23.7 ± 64.3)	1.67 ± 0.11	46.99 (38.3 ± 57.7)	3.36 ± 0.05
*P. xylostella*	30	23.67 (10.9 ± 51.2)	1.02 ± 0.17	49.04 (40.7 ± 58.1)	3.82 ± 0.04
*M. vitrata*	30	25.04 (13.9 ± 45.3)	1.39 ± 0.13	50.80 (41.5 ± 62.2)	3.43 ± 0.05

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
