# Peer review of "Host Immunosuppression Induced by Steinernema feltiae, an Entomopathogenic Nematode, through Inhibition of Eicosanoid Biosynthesis"

_insects, 2019, doi:10.3390/insects11010033_

Round 1

Reviewer 1 Report

The paper is interesting and present quite original data, properly analyzed. Just few notes:

Scientific names, when first reported in a text, must be followed by the Authority and the systematic classification At pag 9 the genetic tree can be improved by adding other species such as Steinernema ichnusae the methodology is not perfectly clear in some points and needs to be better explained

Author Response

Comment #1-1: Scientific names, when first reported in a text, must be followed by the Authority and the systematic classification

Response: All the first scientific names are supplemented with the first author and systematic classification.

Comment #1-2: At page 9 the genetic tree can be improved by adding other species such as Steinernema ichnusae

Response: The species is added to make a new phylogeny tree.

Comment #1-3: the methodology is not perfectly clear in some points and needs to be better explained.

Response: 2.5. section is rephrased as follows: “Median lethal concentration (LC50) was estimated from the mortality data at 72 h after treatment. Median lethal time (LT50) was estimated after treatment with an IJ concentration (80 IJs/larva). To estimate LT50, mortality was assessed every 8 h for 3 days after nematode infection.

Reviewer 2 Report

Ref: Manuscript ID: insects-677745

Miltan Chandra Roy, Dongwoon Lee & Yonggyun Kim: Host Immunosuppression Induced by Steinernema feltiae, an Entomopathogenic Nematode, through Inhibition of Eicosanoid Biosynthesis

This MS deals with nematode infection in three lepidopterans and one coleopteran species. The authors identified a new nematode isolate and studied its pathogenicity to insect hosts. The MS is well written, and I enjoyed reading it. I have only a couple of minor comments on the MS.

Page 1, Abstract, lines 9-10: abbreviate arachidonic acid as AA here. Otherwise, it is not clear to the reader what is AA on line 20.

Page 2, Introduction, the final para: the authors could briefly mention the most important phases of their work here: antibiotic use, PLA2, THC, PO measurements to give the reader a better overview of the methods used, and hypotheses tested. Perhaps, they could cite some previous studies showing the logic of ampicillin, tetracycline, and kanamycin treatment.

Page 6, line 2: “A nematode isolate was collected from Korea”; this is a redundant sentence because this is already stated on P. 3, lines 4-5.

Page 8, Table 1: There is one Steinernema feltiae without specific NCBI-GenBank name. Is this correct? Steinernema feltiae can be shortened as S. feltiae.

Page 12, line 7: “with S. feltiae K1 almost lost their nodule formation”. I would add “ability” after “formation”.

Pages 16-18: Figure 7 A, B, C, D should be moved out of the Discussion to the Results.

Author Response

Comment #2-1: Page 1, Abstract, lines 9-10: abbreviate arachidonic acid as AA here. Otherwise, it is not clear to the reader what is AA on line 20.

Response: It is added.

Comment #2-2: Page 2, Introduction, the final para: the authors could briefly mention the most important phases of their work here: antibiotic use, PLA2, THC, PO measurements to give the reader a better overview of the methods used, and hypotheses tested. Perhaps, they could cite some previous studies showing the logic of ampicillin, tetracycline, and kanamycin treatment.

Response: We added the information as follows: “This study identified a new strain of S. feltiae and analyzed its pathogenicity by assessing insecticidal activity and immunosuppression via measuring PLA2 activity, counting total hemocyte count (THC), and phenoloxidase (PO) activity. It also tested a hypothesis that AA addition could rescue the immunosuppression induced by S. feltiae infection. Finally, it assessed the role of the nematode in suppressing immunosuppression by attenuating symbiotic bacteria by antibiotics.”

Comment #2-3: Page 6, line 2: “A nematode isolate was collected from Korea”; this is a redundant sentence because this is already stated on P. 3, lines 4-5.

Response: It is deleted.

Comment #2-4: Page 8, Table 1: There is one Steinernema feltiae without specific NCBI-GenBank name. Is this correct? Steinernema feltiae can be shortened as S. feltiae.

Response: We revised the Table according the suggestion.

Comment #2-5: Page 12, line 7: “with S. feltiae K1 almost lost their nodule formation”. I would add “ability” after “formation”.

Response: Added

Comment #2-6: Pages 16-18: Figure 7 A, B, C, D should be moved out of the Discussion to the Results.

Response: The figure is moved to Results.

Reviewer 3 Report

The manuscript by Roy et al. explores an important and understudied area of biology; the immunosuppressive activity of EPNs and the relative roles of the nematodes themselves and their bacterial symbionts. These authors are well known in this field of biology and have produced a detailed and careful analysis in the current manuscript. Overall the paper is well-written and the authors are careful in their analyses, there are a few areas where they extend a bit too far in their conclusions, but this can be remedied by textual modifications. It was a pleasure to read this particular paper and I congratulate the authors on another excellent contribution to the field. A more detailed review is written below.

Major Issues:

One area where the authors overextend is in their description and interpretation of the arachidonic acid and dexamethasone experiments. In the introduction the authors suggest that “If nematode contributes to the suppression of insect immune responses by targeting non-PLA2, AA addition would then incompletely rescue the immunosuppression.” This is not necessarily true, the many aspects of insect immunity are inseparably connected. If a large amount of AA is added, this may compensate for other compromised pathways such as AMPs or PO pathways. In the case presented here, the authors use large amounts of AA and DEX in their experiments, such that they should soften their hypothesis that AA addition would only rescue immunosuppression completely if eicosanoid synthesis were the only pathway being targeted. This has not been convincingly demonstrated, though their argument would be much stronger if they had used a lower, more biologically relevant dose of AA in their experiments. How much AA is in an entire larva? 10 micrograms seems excessive and could be more than the whole naïve organism has to begin with. In figure 4B the authors show an effect with only 0.1ug of AA, they should have use this dose in subsequent experiments rather than 10ug; this would have made for a stronger argument that eicosanoid biosynthesis was the only/primary target of immunomodulation. The experiments shown in Figure 5  and 6 use 10ug/larva, which seems quite high compared to what would be physiologically relevant for these larvae. Instead, as discussed in their conclusion, S. feltiae releases a large amount of different proteins (Chang et al. 2019), many of which are likely to target other aspects of immunity such as PO, encapsulation, and AMPs rather than only targeting eicosanoid signaling and biosynthesis.

Figure 1 panels F and G must be missized or mislabeled. S. feltiae males and females are quite different in size, though these images suggest that they are the same size (Nguyen et al. 2007).

In section 3.2 the authors suggest that T. molitor is more tolerant than the other insects they examined. Resistance and tolerance in this context are completely different aspects of immunity and they can only differentiate between them if they have both a measure of health and a measure of the number of IJs that enter the insects (Schneider and Ayres 2008; Schneider 2011; Medzhitov et al. 2012). While the authors have measured mortality, they have not measured IJ invasion and cannot say whether T. molitor simply has fewer IJs that invade or that they are indeed more tolerant to having the same number of IJs. The experiment described exposes all of the insects to the same number of IJs, but the authors do not measure how many of those IJs enter, therefore they cannot differentiate between resistance and tolerance. This language should be modified to address this concern. What they can say is that T. molitor seems less susceptible to infection.

In section 3.5 the authors oversimplify what is likely important in the infection process. They presume that the only potential contributors to pathogenicity are the nematode and its specific bacterial pathogen. They ignore the possibility that in perforating the gut that part of the gut microbiota may be contributing to subsequent septicemia. By using antibiotics to eliminate X. bovienii, they also eliminate any other bacteria that may be present. These experiments would have been better if they had used axenic IJs rather than antibiotics. The authors can correct this by softening their conclusions and adding caveats rather than performing any additional experiments.

In the discussion the authors mention that some of their findings “indicates that the insecticidal activity of S. feltiae is not determined by IJ density only.” When in point of fact they cannot rule this out based solely on their findings since they only measured the density of IJs to which the larvae were exposed rather than the density of IJs that entered the host and participated in infection.

While the authors have included many references and are quite scholarly throughout their manuscript, they have ignored a section of the field on the contribution of the nematode cuticle to immune modulation. There are quite a few papers by Maurizio Brivio’s lab, reviewed last year in Insects (Reviewed in Brivio and Mastore 2018). Perhaps a mention of the possible role of the nematode cuticle in the discussion or introduction would suffice.

References

Brivio, M. F., and M. Mastore, 2018 Nematobacterial Complexes and Insect Hosts: Different Weapons for the Same War. Insects 9.

Chang, D. Z., L. Serra, D. H. Lu, A. Mortazavi and A. R. Dillman, 2019 A core set of venom proteins is released by entomopathogenic nematodes in the genus Steinernema. Plos Pathogens 15.

Medzhitov, R., D. S. Schneider and M. P. Soares, 2012 Disease tolerance as a defense strategy. Science 335: 936-941.

Nguyen, K. B., D. J. Hunt and Z. Mracek, 2007 Steinernematidae: species and descriptions, pp. 121-609 in Entomopathogenic Nematodes: Systematics, Phylogeny and Bacterial Symbionts, edited by K. B. Nguyen and D. J. Hunt. Brill, Boston.

Schneider, D. S., 2011 Tracing personalized health curves during infections. PLoS Biol 9: e1001158.

Schneider, D. S., and J. S. Ayres, 2008 Two ways to survive infection: What resistance and tolerance can teach us about treating infectious diseases. Nat Rev Immunol 8: 889-895.

Author Response

Comment #3-1: One area where the authors overextend is in their description and interpretation of the arachidonic acid and dexamethasone experiments. In the introduction the authors suggest that “If nematode contributes to the suppression of insect immune responses by targeting non-PLA2, AA addition would then incompletely rescue the immunosuppression.” This is not necessarily true, the many aspects of insect immunity are inseparably connected. If a large amount of AA is added, this may compensate for other compromised pathways such as AMPs or PO pathways. In the case presented here, the authors use large amounts of AA and DEX in their experiments, such that they should soften their hypothesis that AA addition would only rescue immunosuppression completely if eicosanoid synthesis were the only pathway being targeted. This has not been convincingly demonstrated, though their argument would be much stronger if they had used a lower, more biologically relevant dose of AA in their experiments. How much AA is in an entire larva? 10 micrograms seems excessive and could be more than the whole naïve organism has to begin with. In figure 4B the authors show an effect with only 0.1ug of AA, they should have use this dose in subsequent experiments rather than 10ug; this would have made for a stronger argument that eicosanoid biosynthesis was the only/primary target of immunomodulation. The experiments shown in Figure 5  and 6 use 10ug/larva, which seems quite high compared to what would be physiologically relevant for these larvae. Instead, as discussed in their conclusion, S. feltiae releases a large amount of different proteins (Chang et al.2019), many of which are likely to target other aspects of immunity such as PO, encapsulation, and AMPs rather than only targeting eicosanoid signaling and biosynthesis.

Response: No information is available in the content of AA during infection. Indeed, little AA is present in naïve larvae of S. exigua. Recent study (Hasan et al., 2019) showed a de novo synthesis of AA from linoleic acid with elongase and desaturase. Thus, during infection, the synthetic rate might control the amount of AA in S. exigua. Fig, 4B showed that at least 10 ug of AA is required for inducing recovery of immunity. Thus, we used all the subsequent assays with 10 ug of AA.  

Comment #3-2: Figure 1 panels F and G must be missized or mislabeled. S. feltiae males and females are quite different in size, though these images suggest that they are the same size (Nguyen et al. 2007).

Response: The scale bars are re-adjusted.

Comment #3-3: In section 3.2 the authors suggest that T. molitor is more tolerant than the other insects they examined. Resistance and tolerance in this context are completely different aspects of immunity and they can only differentiate between them if they have both a measure of health and a measure of the number of IJs that enter the insects (Schneider and Ayres 2008; Schneider 2011; Medzhitov et al. 2012). While the authors have measured mortality, they have not measured IJ invasion and cannot say whether T. molitor simply has fewer IJs that invade or that they are indeed more tolerant to having the same number of IJs. The experiment described exposes all of the insects to the same number of IJs, but the authors do not measure how many of those IJs enter, therefore they cannot differentiate between resistance and tolerance. This language should be modified to address this concern. What they can say is that T. molitor seems less susceptible to infection.

Response: We agree on the concern raised by reviewer. We added the concerns in the discussion as follows: “However, we cannot avoid the pathogenic contribution of the gut microbiota in the host insects, which might infect the insect hemocoel during penetration of IJs from gut lumen. To clarify the role of nematode on the immunosuppression, use of axenic IJs might be optimal in a future study rather than use of antibiotics to kill the symbiotic bacteria.”.

Comment #3-4: In section 3.5 the authors oversimplify what is likely important in the infection process. They presume that the only potential contributors to pathogenicity are the nematode and its specific bacterial pathogen. They ignore the possibility that in perforating the gut that part of the gut microbiota may be contributing to subsequent septicemia. By using antibiotics to eliminate X. bovienii, they also eliminate any other bacteria that may be present. These experiments would have been better if they had used axenic IJs rather than antibiotics. The authors can correct this by softening their conclusions and adding caveats rather than performing any additional experiments.

Response: We agree on the concern raised by reviewer. We changed the sentence as follows: “”.

Comment #3-5: In the discussion the authors mention that some of their findings “indicates that the insecticidal activity of S. feltiae is not determined by IJ density only.” When in point of fact they cannot rule this out based solely on their findings since they only measured the density of IJs to which the larvae were exposed rather than the density of IJs that entered the host and participated in infection.

Response: We agree on the concern raised by reviewer. We changed the sentence as follows: “This indicates that the insecticidal activity of S. feltiae K1 is not determined by IJ density only if we knew the number of the infected IJs within the target insects.”.

Comment #3-6: While the authors have included many references and are quite scholarly throughout their manuscript, they have ignored a section of the field on the contribution of the nematode cuticle to immune modulation. There are quite a few papers by Maurizio Brivio’s lab, reviewed last year in Insects (Reviewed in Brivio and Mastore 2018). Perhaps a mention of the possible role of the nematode cuticle in the discussion or introduction would suffice.

Response: We added the nematode cuticle function in immunosuppression as follows: “In addition, the nematode cuticle plays a crucial role in avoiding insect immune surveillance by disguising nematode surface [49]. Thus, several chemical and physical factors derived from S. feltiae K1 might be responsible for its immunosuppression and insecticidal activity.”

Brivio, M. F., and M. Mastore, 2018 Nematobacterial Complexes and Insect Hosts: Different Weapons for the Same War. Insects 9.
